# Farm living and risk of asthma, atopic eczema, respiratory and food allergy: protocol for a systematic review and meta-analysis

Johnstone Kuya [ORCID],[1] Ogban Omoronyia,[2] Theopista Fokukora,[3]
Aduroja Posi Emmanuel,[4] Olusegun Sunday Ewemooje,[5] Adewumni O Soyege,[6]
Valery Ngo,[7] Felix Emeka Anyiam,[8] Josephine Akua Ackah,[9] Linus Ossai-Chidi,[10]
Beatriz Manuel,[11] Ogechukwu Emmanuel Okondu [ORCID],[6] Bright I Nwaru[12]

For numbered affiliations see end of article.

**Correspondence to**
Dr Bright I Nwaru;
bright.nwaru@gu.se

## ABSTRACT

**Introduction** Asthma and allergic disorders are of global concern and risk for their development results from the interplay of genetic and environmental factors. Among numerous environmental factors identified to influence the disease risk, the role of exposure to a farming environment has gained interest in recent times, but the underlying evidence is conflicting. The review aims to comprehensively identify, critically appraise and synthesise evidence from studies investigating the association between farm living and risk of asthma and allergic diseases in children and adults.

**Methods and analysis** We will identify relevant analytical observational epidemiological studies, including cross-sectional, case-control and cohort studies, by searching PubMed, Cochrane Library, Google Scholar, Cumulative Index to Nursing and Allied Health Literature (CINAHL), WHO Global Health Library, Web of Science, Scopus and Embase. Screening of identified records, data extraction from eligible studies and risk of bias asssssment of eligible studies will be independently undertaken by two reviewers, with arbitration by a third reviewer. The Effective Public Health Practice Project will be employed for the risk of bias assessment. Estimates from studies judged to be clinically, methodologically and statistically homogeneous will be synthesised using random-effects meta-analysis. Heterogeneity will be assessed using the I-squared statistic. We will consider objectively measured or self-reported asthma, atopic dermatitis/eczema, allergic rhinitis, wheeze, and food allergy as primary outcomes.

**Ethics and dissemination** As this study is based solely on the published literature, no ethics approval is required. The study findings will be presented at scientific meetings related to the field of asthma and allergy and will be published in an international peer-reviewed scientific journal.

**PROSPERO registration number** CRD42020208805.

## Strengths and limitations of this study

► Findings from this systematic review will provide a comprehensive and most contemporaneous synthesis of the underlying evidence linking exposure to a farming environment to risk of asthma and allergic diseases since the latest publication of the previous reviews was 5 years ago.

► This systematic review protocol follows the Preferred Reporting Items for Systematic Review and Meta-Analysis Protocols guidelines.

► The identification of studies from leading medical and public health databases, with no geographical or language limitations, will advance the import of this evidence synthesis across settings.

► We anticipate that the definition of farm exposure and asthma and allergic outcomes will vary between studies, thus posing a challenge in deriving pooled estimates of effect.

## BACKGROUND

Globally, there is high morbidity due to asthma and allergic diseases.[1] This group of immunological conditions have continued to contribute significantly to the burden of disease and years lost due to disability in diverse global settings.[1] Currently, an estimated 339 million people have asthma, which represents the most common chronic disease among children.[2] The development of asthma and allergic diseases results from the interplay of genetic and environmental factors.[3]

Numerous environmental factors linked to the risk of asthma and allergic diseases have been identified, among which include household exposures (eg, dust mites, household dampness, indoor air pollution, moulds and domestic pets) and non-household exposures (eg, respiratory infections, tobacco smoke, outdoor air pollution and exposure to a farming environment).[4] However, findings from available studies on the influence of these environmental factors are

conflicting.[5–7] With a specific focus on the role of the farming environment, over the last three decades, several studies have reported protection for atopy, allergic disease and asthma,[8] while some studies have also reported that exposure to the farming environment may increase the risk of the onset of the diseases.[9 10] While most of these studies have focused on exposure to a farming environment during early childhood and asthma and allergy in childhood, there are also some studies in adults.

The amount and variety of microbial exposure from farm living are thought to be a key mechanism through which the farming environment influences the risk of asthma and allergies.[11] Greater microbial exposure, characteristic of the more unhygienic farm environment, stimulates the development of a more effective immune system.[12] Microbial exposure helps the immune system to learn to efficiently avoid over-reaction to potentially harmless substances that may have been perceived as allergens.[11] However, the effect of microbial exposure may differ according to farm living environment or farming activities.[13] A large-scale multicentre cross-sectional study among rural school children in five European countries found that pig farming, haying, consumption of farm milk and dwelling in the farm were associated with substantially reduced risk of asthma compared with wheeze.[14] The high proportion of whey protein found in farm milk is thought to initiate the process of more effective regulation of the immune system to prevent over-reactivity to diverse substances.[15] Furthermore, sialic acid, more commonly found in farm animals and their products, is thought to contribute to antibody-mediated regulation of inflammatory reactions to substances.[16]

Several systematic reviews have been published that summarised the available evidence linking the farming environment to the risk of asthma and allergy. Dicks *et al*[17 18] in their review revealed that habitation of children in an area with intensive farming was associated with an increased risk of being asthmatic. The review considered asthma and wheeze as the only outcomes, but importantly, it was based on searches made only in PubMed and was undertaken by only one person. Without consideration of other allergic outcomes other than asthma and wheeze does not give a comprehensive overview of the underlying evidence. Searching only one database of several databases that collect scientific research also indicates that the generated evidence is limited, given that the overlap between PubMed and other research databases is only partial. A systematic review should provide the highest evidence on a research topic and therefore the process needs to be transparent and reproducible, with at least two reviewers going through the processes of the review being a mandatory requirement; however, Genuneit[19] failed in this aspect as it was undertaken only by one person. The systematic review by Campbell *et al*[20] focused only on objective measures of atopy (based on skin prick test (SPT) and immunoglobulin E (IgE) measurements) and was also based on studies retrieved only from PubMed. Similar limitations affected this review, in that it investigated a small subset of atopic outcomes and performed a limited search of available literature. Including both objective measures and subjective measures in the assessment of allergy, which are heterogeneous in nature, will provide a clearer insight into the impact of farm living on these outcomes.

To gain a clearer and comprehensive appreciation of the underlying evidence linking exposure to a farming environment to risk of asthma and allergic diseases, the noted limitations in previous systematic reviews need to be overcome: there should be an exhaustive set of outcomes included; there should be an exhaustive search of the literature; and the review processes should be transparent and reproducible, with at least two people doing the review. Moreover, since the publication of the previous reviews, the latest published 5 years ago, there have been several studies now published on the topic. These studies need to be integrated into the evidence synthesis to provide a contemporaneous synthesis of the evidence base.

## AIMS AND OBJECTIVES

By overcoming limitations of previous systematic reviews, including limited outcomes studied, limited databases search, sole reviewer, and a long time since their conduct, we aim to comprehensively identify, critically appraise and synthesise evidence from studies investigating the association between farm living and risk of asthma and allergic diseases in children and adults.

Specific objectives are to synthesise the evidence on:
1. The association between farm living and the risk of the onset of asthma and allergic diseases in children and adults.
2. The association between farm living and the risk of outcomes (eg, indicators of exacerbation and severity) of asthma and allergic diseases in children and adults with already established disease.
3. Whether the association between farm living and risk of asthma and allergy depends on the type of farming environment (eg, pig farmers vs cattle rearers, small vs large-scale farmers).
4. Whether the association between farm living and risk of asthma and allergy differs between geographical regions of the world (eg, high-income vs low-income and middle-income countries).

## METHODS
### Study eligibility criteria
Type of studies: we will include analytical observational epidemiological studies (cross-sectional, case-control and cohort studies). We will exclude clinical case studies, case-series discussion papers, non-research letters and editorials, and animal studies.

## Types of participants

We will include studies that included participants of any age, in which the association between farm living and asthma and allergic diseases have been investigated, regardless of gender, age and nationality. Children will be defined as those below the age of 18 years and adults as those 18 years and above.

## Types of exposure

Studies that have investigated the role of farm living, regardless of the type of farm environment and the place of farming (rural vs urban). We will not consider the specific constituents of the farm (eg, bioaerosols) and we will define farm living as having a residence in a farming environment, either as an active farmer or passive farmer (ie, living on a farm but not a farmer by yourself).

## Types of outcome measures

As primary outcomes, we will consider objectively-measured or self-reported asthma, atopic dermatitis/eczema, allergic rhinitis, wheeze and food allergy. Our secondary outcomes will include atopic sensitisation measured by SPT or antigen-specific IgE; indicators of disease severity and measures of quality of life; asthma exacerbations; use of medications for respective allergic diseases and asthma; hospitalisation for asthma; indicators of airway function including peak expiratory flow, forced expiratory volume in 1s, forced vital capacity, forced expiratory flow rate, or other pulmonary function tests (oscillometry or exhaled nitric oxide analysis)).

## Information sources

### Database searches

Relevant studies/articles from the onset of the following electronic databases until will be sought from: PubMed, Cochrane Library, Google Scholar, CINAHL, WHO Global Health Library, Web of Science, Scopus and Embase. There will be no restriction on language, and we will endeavour to translate papers not published in English. We will supplement the records retrieved from the respective databases by contacting previous authors on the topic to identify any studies our searches might have missed.

### Search strategy

Search strategies covering concepts and synonyms of farm living and the study outcomes (refer to the online supplemental apendix of search strategies) will be adapted to search the other databases.

### Screening of retrieved literature

The literature retrieved from the databases will be transferred to Endnote for removal of duplicate papers; thereafter the papers will be exported to Rayan for further screening. Two reviewers will separately evaluate the titles and abstracts of the deduplicated articles based on the study inclusion and exclusion criteria. The full texts of eligible articles will be collected and further critically reviewed by the two reviewers for inclusion.

Any differences between the two reviewers in the evaluation processes will be settled by consensus, and a third reviewer will arbitrate any disagreement. The Preferred Reporting Items for Systematic Reviews and Meta-Analyses (PRISMA) flow chart will be used to report the screening process.[21] The (Meta-analyses Of Observational Studies in Epidemiology) MOOSE guideline will be used in reporting the results from the meta-analyses.[22]

## Registration and reporting

This study protocol has been registered with the University of York Centre for Reviews and Dissemination International prospective register of systematic reviews (PROSPERO number CRD42020208805). The protocol is reported according to the PRISMA-P guidelines for reporting of systematic review protocols, awaiting registration number.

## Data extraction

A standardised data extraction form will be developed and will be used by at least two members of the review team to independently extract relevant study data from the full-text papers included in the systematic review. Before full use with all included studies, the data extraction form will be first piloted using a couple of the studies. Following the piloting, necessary amendments as required will be made to the extraction form and thereafter will be used for data extraction of all the studies. The data to be extracted will include key characteristics of the included studies and study participants (eg, age, gender, family history of allergy/asthma, etc), study setting, study design, details of exposures, outcomes, statistical approach to the analysis of study data, missing data and details of study findings. Author(s) of included studies will be contacted to request additional data where such data are missing from the studies. Any discrepancies between reviewers in the data extraction will be resolved by either consensus or a third reviewer will arbitrate.

## Risk of bias in individual studies

The risk of bias in individual studies will be undertaken by two independent reviewers using the Effective Public Health Practice Project (EPHPP) quality assessment tool. The EPHPP tool has six domains used to assess the potential risk of bias in each study, including selection bias, study design, confounders, blinding, data collection method, and withdrawals and dropouts. Each of these domains is rated as strong, moderate or weak. Based on this domain rating, a global rating that integrates all the domain ratings is determined for each study. Any discrepancies between reviewers in the risk of bias assessment will be resolved by either consensus or a third reviewer will arbitrate.

## Data synthesis

Descriptive data from all studies will be presented in a table of characteristics. A narrative synthesis of the collected data across all studies will be undertaken. In addition, we will perform meta-analysis using random-effects models

for studies judged to be sufficiently homogenous with regards to a similarity in methods, study population, study design, exposure measures and assessment, and outcomes and assessment. Heterogeneity will be assessed using the I-squared statistic. We will undertake subgroup analysis to explore any potential differences in the estimates of the association between farm living and the study outcomes by age (children vs adults), world region (high income vs low-income and middle-income countries) and setting of the farm (rural vs urban). With a sufficient amount of studies, we will perform meta-regression to further explore the underlying reasons for any potential heterogeneity in estimates between studies. Sensitivity analysis will be performed to explore any potential scenario that can change the conclusion of our findings, for example, by excluding all low-quality studies from the meta-analysis and evaluate whether the results from high-quality studies differ from all studies included together. Publication bias will be explored using funnel plots and by calculating the Begg and Egger test.[23 24] The meta-analysis will be undertaken in RevMan (Review Manager V.5.4). All this will be expressed with a 95% CI.

## Ethics and data management plan

As the study involves only the collection and analysis of data from already published literature, no ethics approval is required. All study data will be handled only by the researchers involved in the study and if relevant will be made open in the journal where the systematic review results will be published.

## Patient and public involvement

No patient or members of the public was involved in the conception of this study and the development of the study protocol.

## Dissemination

The study findings will be presented at scientific meetings related to the field of asthma and allergy and will be published in an international peer-reviewed scientific journal.

**Author affiliations**
[1]Public Health, Africa Development and Strategy Centre, Luanda, Vihiga, Kenya
[2]Community Medicine, University of Calabar, Calabar, Nigeria
[3]Department of Public Health, Cavendish University, Kampala, Uganda
[4]Department of Health Promotion and Education, University of Ibadan, Ibadan, Nigeria
[5]Department of Statistics, Federal University of Technology Akure, Akure, Nigeria
[6]Department of Public Health, Babcock University, Ilishan-Remo, Ogun, Nigeria
[7]Department of Global Health, University of Gothenburg, Goteborg, Sweden
[8]Centre for Health and Development, University of Port Harcourt, Choba, Nigeria
[9]Department of Population and Health, University of Cape Coast, Cape Coast, Ghana
[10]Department of Medical Microbiology, University of Port Harcourt Teaching Hospital, River state, Nigeria
[11]Department of Community Health, Eduardo Mondlane University, Maputo, Mozambique
[12]Krefting Research Centre, Institute of Medicine, Sahlgrenska Academy, University of Gothenburg, Goteborg, Sweden

**Contributors** All authors conceived of the presented idea of the study and supervised the project. JK and OO wrote the first draft of the manuscript with TF, APE, OSE and AOS. OSE, APE, FEA, OEO with assistance from VN, BM, LO-C and JAA provided methodological and statistical expertise. All authors contributed to subsequent drafts, provided critical feedback, refinement of the study protocol and approved the final manuscript. BIN is the guarantor of the paper.

**Funding** The authors have not declared a specific grant for this research from any funding agency in the public, commercial or not-for-profit sectors.

**Competing interests** None declared.

**Patient consent for publication** Not applicable.

**Provenance and peer review** Not commissioned; externally peer reviewed.

**ORCID iDs**
Johnstone Kuya http://orcid.org/0000-0001-8458-1655
Ogechukwu Emmanuel Okondu http://orcid.org/0000-0003-0872-7581

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
