## [Reviewer comments · BMJ Open]

ARTICLE DETAILS

TITLE (PROVISIONAL)	Farm living and risk of asthma, atopic eczema, respiratory and food allergy: protocol for a systematic review and meta-analysis.
AUTHORS	Kuya, Johnstone; Omoronya, Ogban; Fokukora, Theopista; Emmanuel, Aduroja Posi; Ewemooje, Olusegun; Soyege, Adewumni; Ngo, Valery; Anyiam, Felix Emeka; Ackah, Josephine Akua; Ossai-Chidi, Linus; Manuel, Beatriz; Okondu, Ogechukwu; Nwaru, Bright

VERSION 1 – REVIEW

REVIEWER	Nurmatov, Ulugbek Division of Population Medicine, School of Medicine, Cardiff University
REVIEW RETURNED	03-Mar-2021

GENERAL COMMENTS	Overall, the manuscript is well-written, methodologically rigorous and reads well. My only concern is how to deal with ongoing studies, whether or not to contact international experts in this field of research? After addressing these questions the manuscript can be published.
---

REVIEWER	Luan, Chu University of Saskatchewan
REVIEW RETURNED	12-May-2021

GENERAL COMMENTS	1. The authors should consider changing the title to reflect the main outcomes of the systematic review. It's beyond asthma and allergy. The authors also included food allergy, which requires a lot of work. Consider narrowing the field of search: respiratory allergy, asthma, food allergy, etc. in separate papers.2. Consider the heterogeneity of outcome definitions (especially atopy by skin tests and IgE tests), will the meta-analyses be valid?3. What are the age ranges for children and adults?4. Does the author consider occupational asthma?5. What makes this systematic review different from previous/current reviews?
---

REVIEWER	Lawson , Joshua A. University of Saskatchewan
REVIEW RETURNED	22-May-2021

GENERAL COMMENTS	This was a relatively well written and well thought out protocol for a systematic review that will look at the relationship between farm living and asthma and allergic outcomes. While this could be an important review in the future, I have a few comments to help the authors.  1. The methods for the review are relatively strong, following textbook protocols set out for reviews. The authors have also registered the review with Prospero. As such, the work could be found by others looking to see what is out there and standard review methods, common to most who plan a systematic review, were planned. 2. The purpose of a systematic review is to be more narrow in scope. However, the objectives are relatively broad and move beyond onset of asthma and allergic conditions and move into severity and other indicators. These could be split into two reviews. Related, it was mentioned that one of the limitations of Campbell et al was that objective measures of atopy were used. However, this is a strength as it focuses on a strong outcome that is well defined. 3. The primary objective looks at the risk of onset of asthma and allergic diseases, which implies temporality. However, the only design that can explicitly look at temporality in the current methodology is the cohort study. In this field, there are a relatively low number of cohort studies and a large number of cross-sectional studies. How will the objective be met when cross-sectional studies cannot be used to look at temporality and, therefore, onset? 4. When looking at the exposures, farm living is the exposure of interest. This should be defined. What is meant by farming, especially when the authors consider urban farming (how many studies are based on urban farms?). 5. There is a large list of primary outcomes and a very long list if outcomes in general. Included in this is food allergy, which is novel but can be unique compared to the other conditions listed. Is there evidence of a relationship between food allergy and farming. Also, will the outcomes be considered as a composite outcome or considered separately in the analyses? It would be advisable to look at them separately since the mechanisms between the exposures and outcomes can vary. 6. I am pessimistic about running a meta-analysis as there is a very large degree of heterogeneity in exposures and outcomes in this field. 7. The EPHPP is a good tool but in this situation, almost all the studies will be classified as low strength due to lack of blinding and lack of randomization. As such, this type of sensitivity analysis (remove weak studies) will be difficult.
---

VERSION 1 – AUTHOR RESPONSE

Reviewer: 1

Dr. Ulugbek Nurmatov, Division of Population Medicine, School of Medicine

Comment:

Overall, the manuscript is well-written, methodologically rigorous and reads well.

My only concern is how to deal with ongoing studies, whether or not to contact international experts in this field of research? After addressing these questions the manuscript can be published.

Response:

We will not consider ongoing studies in this review but will contact previous authors on the topic to see if our databases searches might have missed any relevant studies. We have amended the relevant section on the manuscript (page 6, lines 209-211).

Reviewer: 2

Dr. Chu Luan, University of Saskatchewan

Comment:

1. The authors should consider changing the title to reflect the main outcomes of the systematic review. It's beyond asthma and allergy. The authors also included food allergy, which requires a lot of work. Consider narrowing the field of search: respiratory allergy, asthma, food allergy, etc. in separate papers.

Response:

We thank the reviewer for the suggestion. We have now amended the title of the manuscript as suggested. We believe that presenting the evidence for a comprehensive set of outcomes on the topic will provide a clearer picture and allow a comparison of where the evidence is robust and where it is not. For this reason, we decided to keep all the outcomes we proposed to include in the review.

Comment:

2. Consider the heterogeneity of outcome definitions (especially atopy by skin tests and IgE tests), will the meta-analyses be valid?

Response:

As indicated in our meta-analysis plan (page 7, lines 258-261), we will evaluate how homogeneous the studies are (e.g. in terms of study design, population characteristics, exposure definitions, and outcome definition) before including them in the meta-analyses. Whereas the studies are too heterogeneous to allow deriving pooled estimates, we will use only narrative synthesis to summarize the evidence.

Comment:

3. What are the age ranges for children and adults?

Response:

These have now been defined (page 5, lines 185-186)

Comment:

4. Does the author consider occupational asthma?

Response:

We will include all types of asthma and will explore, where possible if the association differ by asthma types.

Comment:

5. What makes this systematic review different from previous/current reviews?

Response:

We outlined several limitations of previous reviews on the topic (page 4, lines 139-147) and highlighted our aim to overcome these limitations in the current review (page 4, lines 158-161).

Reviewer: 3

Dr. Joshua A. Lawson, University of Saskatchewan

Comments to the Author:

This was a relatively well written and well thought out protocol for a systematic review that will look at the relationship between farm living and asthma and allergic outcomes. While this could be an important review in the future, I have a few comments to help the authors.

Comment:

1. The methods for the review are relatively strong, following textbook protocols set out for reviews. The authors have also registered the review with Prospero. As such, the work could be found by others looking to see what is out there and standard review methods, common to most who plan a systematic review, were planned.

Response:

We thank the reviewer for the positive feedback.

Comment:

2. The purpose of a systematic review is to be narrower in scope. However, the objectives are relatively broad and move beyond the onset of asthma and allergic conditions and move into severity and other indicators. These could be split into two reviews. Related, it was mentioned that one of the limitations of Campbell et al was that objective measures of atopy were used. However, this is a strength as it focuses on a strong outcome that is well defined.

Response:

We agree with the author that the inclusion of objective measures in the study by Campbell and colleagues was a strength, but we note that the heterogeneous nature of allergic outcomes warrants that both objective and subjective measures of the disease should be assessed in order to provide more clearer picture on the topic. We have now amended this part of the manuscript to reflect this (page 4, lines 144-146). We decided to include both outcomes of onset among participants who did not previously have the disease and severity outcomes among participants who already have the disease in order to provide a comprehensive picture of the primary and secondary preventative impact of farm living. From our preliminary assessment, studies looking at the severity of the disease are few, so we can easily handle them in this review.

Comment

3. The primary objective looks at the risk of the onset of asthma and allergic diseases, which implies temporality. However, the only design that can explicitly look at temporality in the current methodology is the cohort study. In this field, there are a relatively low number of cohort studies and a large number of cross-sectional studies. How will the objective be met when cross-sectional studies cannot be used to look at temporality and, therefore, onset?

Response:

We thank the reviewer for raising this important point. We will reflect the quality of each study in the quality appraisal we will conduct, which will include an assessment of the ability of each study to

assess temporality. Study designs that are weak in assessing temporality will be downgraded (receive grading of poor quality) with regards to this aspect.

Comment:

4. When looking at the exposures, farm living is the exposure of interest. This should be defined. What is meant by farming, especially when the authors consider urban farming (how many studies are based on urban farms?).

Response:

We have now defined farm living in the manuscript (page 5, lines 191-192).

Comment:

5. There is a large list of primary outcomes and a very long list of outcomes in general. Included in this is food allergy, which is novel but can be unique compared to the other conditions listed. Is there evidence of a relationship between food allergy and farming? Also, will the outcomes be considered as a composite outcome or considered separately in the analyses? It would be advisable to look at them separately since the mechanisms between the exposures and outcomes can vary.

Response:

We included a comprehensive set of outcomes to ensure that we synthesize the evidence on all possible allergy outcomes that have been investigated in previous studies, including the potential for the association between farm living and food allergy. Our preliminary searches have not shown any association between farm living and food allergy, but several studies have looked at specific allergens based on IgE or SPT measurements and in some studies, these measures are taken as a proxy for food allergy. If there is an association between farm living and food allergy, we aim to uncover and synthesize such in this review. We will analyze each outcome separately and not treat them as a composite outcome.

Comment:

6. I am pessimistic about running a meta-analysis as there is a very large degree of heterogeneity in exposures and outcomes in this field.

Response:

As indicated in our meta-analysis plan (page 7, lines 258-261), we will evaluate how homogeneous the studies are (e.g. in terms of study design, population characteristics, exposure definitions, and outcome definition) before including them in the meta-analyses. Whereas the studies are too heterogeneous to allow deriving pooled estimates, we will use only narrative synthesis to summarize the evidence.

Comment:

7. The EPHPP is a good tool but in this situation, almost all the studies will be classified as low strength due to lack of blinding and lack of randomization. As such, this type of sensitivity analysis (remove weak studies) will be difficult.

Response:

We will adapt the EPHPP tool to this review. In those aspects of the tool, e.g. blinding, that is not relevant for the study designs included in this review will not be used to assess the studies.

Reviewer: 1

Competing interests of Reviewer: None declared

Reviewer: 2

Competing interests of Reviewer: I have no competing interests.

Reviewer: 3

Competing interests of Reviewer: None.

VERSION 2 – REVIEW

REVIEWER	Luan, Chu University of Saskatchewan
REVIEW RETURNED	12-Oct-2021

GENERAL COMMENTS	The revised version is well organized and articulated.
--

REVIEWER	Lawson , Joshua A. University of Saskatchewan
REVIEW RETURNED	08-Oct-2021

GENERAL COMMENTS	I have no additional questions. Thank you.
--